# Genome-Scale Metabolic Reconstruction, Non-Targeted LC-QTOF-MS Based Metabolomics Data, and Evaluation of Anticancer Activity of *Cannabis sativa* Leaf Extracts

**DOI:** 10.3390/metabo13070788

**Published:** 2023-06-24

**Authors:** Fidias D. González Camargo, Mary Santamaria-Torres, Mónica P. Cala, Marcela Guevara-Suarez, Silvia Restrepo Restrepo, Andrea Sánchez-Camargo, Miguel Fernández-Niño, María Corujo, Ada Carolina Gallo Molina, Javier Cifuentes, Julian A. Serna, Juan C. Cruz, Carolina Muñoz-Camargo, Andrés F. Gonzalez Barrios

**Affiliations:** 1Group of Product and Process Design, Department of Chemical and Food Engineering, Universidad de los Andes, Bogotá 111711, Colombia; fd.gonzalez@uniandes.edu.co (F.D.G.C.); ad.sanchez@uniandes.edu.co (A.S.-C.); 2Applied Genomics Research Group Vice-Presidency for Research and Creation, Universidad de los Andes, Bogotá 111711, Colombia; mi.guevara34@uniandes.edu.co; 3Metabolomics Core Facility—MetCore Vice-Presidency for Research and Creation, Universidad de los Andes, Bogotá 111711, Colombia; m.santamariatorres@uniandes.edu.co (M.S.-T.); mp.cala10@uniandes.edu.co (M.P.C.); 4Laboratory of Mycology and Phytopathology (LAMFU), Department of Biological Sciences and Department of Chemical and Food Engineering, Universidad de los Andes, Bogotá 111711, Colombia; srestrep@uniandes.edu.co; 5Leibniz-Institute of Plant Biochemistry, Department of Bioorganic Chemistry, Weinberg 3, 06110 Halle, Germany; miguelangel.fernandeznino@ipb-halle.de; 6Ecomedics S.A.S., Commercially Known as Clever Leaves, Calle 95 # 11A-94, Bogota 110221, Colombia; mariacorujobesga@gmail.com; 7Chemical and Biochemical Processes Group, Department of Chemical and Environmental Engineering, National University of Colombia, Bogotá 11001, Colombia; acgallom@unal.edu.co; 8Research Group on Nanobiomaterials, Cell Engineering and Bioprinting (GINIB), Department of Biomedical Engineering, Universidad de los Andes, Bogotá 111711, Colombia; jf.cifuentes10@uniandes.edu.co (J.C.); ja.serna10@uniandes.edu.co (J.A.S.); jc.cruz@uniandes.edu.co (J.C.C.); c.munoz2016@uniandes.edu.co (C.M.-C.)

**Keywords:** *Cannabis sativa*, plant genome-scale metabolic reconstruction, metabolomic validation, secondary metabolism, anticancer activity

## Abstract

Over the past decades, Colombia has suffered complex social problems related to illicit crops, including forced displacement, violence, and environmental damage, among other consequences for vulnerable populations. Considerable effort has been made in the regulation of illicit crops, predominantly *Cannabis sativa*, leading to advances such as the legalization of medical cannabis and its derivatives, the improvement of crops, and leaving an open window to the development of scientific knowledge to explore alternative uses. It is estimated that *C. sativa* can produce approximately 750 specialized secondary metabolites. Some of the most relevant due to their anticancer properties, besides cannabinoids, are monoterpenes, sesquiterpenoids, triterpenoids, essential oils, flavonoids, and phenolic compounds. However, despite the increase in scientific research on the subject, it is necessary to study the primary and secondary metabolism of the plant and to identify key pathways that explore its great metabolic potential. For this purpose, a genome-scale metabolic reconstruction of *C. sativa* is described and contextualized using LC-QTOF-MS metabolic data obtained from the leaf extract from plants grown in the region of Pesca-Boyaca, Colombia under greenhouse conditions at the Clever Leaves facility. A compartmentalized model with 2101 reactions and 1314 metabolites highlights pathways associated with fatty acid biosynthesis, steroids, and amino acids, along with the metabolism of purine, pyrimidine, glucose, starch, and sucrose. Key metabolites were identified through metabolomic data, such as neurine, cannabisativine, cannflavin A, palmitoleic acid, cannabinoids, geranylhydroquinone, and steroids. They were analyzed and integrated into the reconstruction, and their potential applications are discussed. Cytotoxicity assays revealed high anticancer activity against gastric adenocarcinoma (AGS), melanoma cells (A375), and lung carcinoma cells (A549), combined with negligible impact against healthy human skin cells.

## 1. Introduction

Over the past decades, Colombia has suffered from complex social problems related to illicit crops, including forced displacement, violence, and environmental damage, among other consequences for vulnerable populations [1]. Considerable effort has been made in Colombia to address this issue by creating a regulatory framework for import, export, cultivation, extraction, and research activities, especially of *Cannabis sativa* [2,3]. When the contingency caused by the coronavirus began, the former Minister of Health authorized Resolution 315 of 2020, which updates the lists of precursor drugs subject to state control and gives free access to the sale of master formulations (preparations made for medical indications) in order to eliminate some access barriers for research, medical, and scientific use [4]. In addition, two years later, Resolution 227 of 2022 was approved, regulating the use of medicinal *C. sativa* (non-psychoactive components) in food, beverages, and dietary supplements. Furthermore, since the beginning of this year, the national government, through Resolution 2808 of 2022, decided to include magistral preparations of *C. sativa* medicines within the health benefits plan for patients with pathologies such as refractory epilepsy, fibromyalgia, sleep and appetite disorder, cachexia due to cancer, insomnia, chronic pain, neuropathic pain, and pain associated with cancer, in order to address those public health concerns [5]. These laws laid the groundwork for the cultivation of *C. sativa* plants, the emergence of the medical cannabis industry, and safe access to medical and scientific use, among other developments. Hence, the current regulatory framework promotes scientific knowledge of *C. sativa* and allows for the exploration of potential markets for its alternative uses [6].

The field of research related to *C. sativa* has been expanding at an accelerated rate [7] thanks to the biotechnological capacity hidden in the plant. It is estimated that *C. sativa* can produce approximately 750 specialized secondary metabolites [8,9,10]. Some of the most relevant are monoterpenes, sesquiterpenoids, triterpenoids, essential oils, flavonoids, phenolic compounds (known as polyphenols [7]), lignans, stilbenoid derivatives, alkaloids, amino acids, spiro-indans, steroids, and glycoproteins, mainly due to their anticancer properties [8,11,12,13,14]. Previous studies have shown a synergy among the metabolic compounds of the plant that, as a whole, show different behavior compared to the individual performance of each metabolite due to the “entourage effect” [15,16]. It is established that *C. sativa* chemotypes’ rich cannabinoid and terpenoid content offer better pharmacological activities that are able to broaden clinical applications and improve therapeutic issues [17,18,19]. In the same way, remarkable anticancerogenic activity has been demonstrated for *C. sativa* extracts against different carcinoma cell lines such as melanoma [20], ovarian [21], prostate [22], breast [23], and pancreatic cancer [16]. These studies have revealed a reduction in tumor growth and promotion of apoptosis and autophagy in carcinoma cells [15,23,24,25]. At the taxonomic level, chemotypes are grouped in terms of the relative amounts of their main compounds, the cannabinoids. Drug-type plants (chemotype I) contain high concentrations of the most prevalent cannabinoid known for its psychotropic capacity, (-)-trans-∆9-tetrahydrocannabinol, or D9-THC. When the cannabinoid content corresponds mostly to the second most abundant cannabinoid in the *C. sativa* plant, cannabidiol, CBD, it corresponds to chemotype III [26]. Finally, chemotype II, which is very scarce, is defined as a balanced content of the two main cannabinoids [27].

For all these reasons, it is critical to understand plant metabolism on a system-wide level to identify metabolic pathways involved in the production of key metabolites, characterize specific phenotypes influenced by environmental factors, and explore alternative uses of the leaf, such as nutraceuticals.

In the last two decades, Genome-Scale Metabolic (GEM) reconstructions have become a fundamental tool taking advantage of the development of high throughput data of omics technologies to study and understand the complex interactions of organisms [28]. Regarding the development of omics technologies in *C. sativa,* the first sequenced and assembled genome was produced in 2011 by Grassa et al. [29] and since then, publications based on whole-genome sequencing and population studies [30,31,32], transcriptomics [33], proteomics [34], and metabolomics have resolved compelling questions about the chemotype of the plant and its relationship with geography or characteristic markers [9]. Additionally, studies of *C. sativa* on the metabolic response of the plant under different degrees of stress [35], its potential uses in different industries [14,16], and particularly nutraceuticals [36,37], such as evaluation of anti-malarial activity [38], observation of in vivo antioxidant effects [39], and pathogen resistance [30], among others, stand out.

Meanwhile, in plant systems biology, genome-scale modeling has advanced considerably thanks to the reconstruction of *Arabidopsis thaliana*, *Zea mays, Oryza sativa,* and *Saccharum officinarum*, among others [40], which have proven accurate predictions focused on specific aspects of central carbon metabolism. For *Arabidopsis thaliana,* GEM modeling has evolved from the production of biomass components observed in experimental data to the inclusion of compartments (cytosol, plastid, mitochondrion, peroxisome, and vacuole), calculation of cell maintenance energy costs, description of photosynthetic processes, integration of secondary metabolism pathways, gene expression, proteomic data, and multi-tissue models [41]; as an example, Scheunemann et al. used the Plant SEED scheme to obtain reconstructions and subsequently integrate transcriptomics data extracted from different plant tissues [42].

Here we present a Genome-Scale Metabolic (GEM) reconstruction of *C. sativa* with an analysis of non-targeted LC-QTOF-MS (Liquid Chromatography-Quadrupole Time-of-Flight Mass Spectrometry)-based metabolomics data and evaluation of cytotoxicity and anticancer activity of leaf extracts, which could help to pave the way for the development of alternative uses of the leaf with potential applications in the food, cosmetic, textile, and agrochemical industries and also to enhance exploration of anticancer, analgesic, and anti-inflammatory compounds [11]. To our knowledge, this is the first attempt to comprehensively describe the metabolic capacities of *C. sativa* leaf (including both primary and secondary metabolism) based on a Genome-Scale Metabolic reconstruction; the contextualization of the reconstruction was carried out via LC-QTOF-MS to favor the identification of metabolites with known (anticancer, due to cannabinoids) and alternative properties (nutraceuticals, due to flavonoids and amino acids) [43].

## 2. Materials and Methods

### 2.1. Metabolic Reconstruction

A description of the metabolic reconstruction workflow is shown in Figure 1. First, the reference genome reported by Grassa was downloaded from NCBI [44]. The size of the genome is reported to be 875.7 Mb, along with a level of assembly up to the chromosomes. The annotation reports 31,170 genes, 25,296 of which are protein-coding genes [44].

Two processes were carried out with the data: functional annotation and automatic reconstruction of the metabolite network (Figure 1).

#### 2.1.1. Functional Annotation and Automated Reconstruction

Starting from the updated version of the *C. sativa* reference genome annotation [44], functional annotation and automated draft reconstruction were conducted based on the Plant SEED workflow for GEM [45,46]. The Plant SEED database (licensed under a Creative Commons Attribution 4.0 International License) describes the core metabolism of the plants and includes several refinement reconstruction steps such as embodiment of reaction stoichiometry and directionality, compartmentalization, transport reactions, charged molecules, and proton balancing on reactions, among others [28,46].

Next, the conversion of reconstructed data into a computable format was performed using the COBRA toolbox (GNU General Public License) and loading the reconstruction into MATLAB (Licence number 40902167) [47,48]; the topological metrics were obtained to evaluate the stoichiometric matrix, and an objective function was set based on the biomass composition of the plant cell [49].

#### 2.1.2. Refinement of Reconstruction

After the first draft model was obtained, a great deal of work was required until the model represented the phenotypic states of the organism [50]. Gap-find was utilized to identify network pathologies which include root no-consumption, root no-production, downstream no-production, and upstream no-consumption and blocked reactions [28,51].

##### Identify Candidate Reactions to Fill Gaps

An exhaustive review of the literature was carried out to identify reactions related to the secondary metabolism of the plant that could fill the gaps and facilitate integrating diverse metabolic pathways taking place in the different cellular compartments [6,11,14,16,52,53,54,55]. Furthermore, KEGG tools were used to complement the metabolic information of the reconstruction through a second functional annotation carried out based on BlastKOALA (KEGG Orthology and Links Annotation) [56]. This was aided by the updated annotation release of the *C. sativa* reference genome [44].

##### Add Gap Reactions to Reconstruction

Regarding the manual curation of models, one of the most complex problems researchers face is the diversity of terminology in reference databases. The present model relies on the Model SEED repository [57], which involves several databases (KEGG, MetaCyc, AraGEM, BiGG, Maize_C4GEM, PlantCyc, and TS_Athaliana, among others) and adds a unique identifier to them [12].

An iterative workflow was carried out to add reactions identified previously to the reconstruction. First, reactions were transformed to the ModelSEED nomenclature taking into account reference ModelSEED database information. Next, renamed reactions were integrated into the reconstruction, considering each compartment of the reconstruction. Finally, a network evaluation was carried out, looking for additional gaps that could be generated for new reactions (Figure 1).

Once the reconstruction was obtained, successive flux balance analyses (FBAs) were carried out. FBA is a mathematical approach that calculates the flow of metabolites through metabolic reconstruction, making it possible to predict the growth rate of an organism [58]. This is done by taking advantage of the constraints imposed by the stoichiometric coefficients of each reaction in the metabolic fluxes. FBAs of *C. sativa* are based on the biomass composition of the plant cell [49] as the objective function of the model (Appendix A) and then evaluating the flow distribution within the system.

### 2.2. Chromatographic Analysis of C. sativa Leaf: LC-PDA and RP-LC-QTOF-MS

#### 2.2.1. Plant Material and Extraction

The sample material was obtained from plants grown in the region of Pesca-Boyaca, Colombia, under greenhouse conditions at the Clever Leaves facility, in a legal operation and under controlled growing conditions, following the guidelines for good agricultural and collection practices (GACP) for starting materials of herbal origin.

The drying process of the plant material was carried out in rooms with controlled conditions for this purpose. The extraction process was carried out from fresh leaf tissue that was ground to a particle size of 1.4 mm, at a 5:1 ratio of ethanol to dry leaves by weight. Constant agitation was performed in a Heidolph shaker at 2000 rpm for 4 h. The supernatant was transferred to a new vial.

Subsequently, the extract obtained was used for LC-PDA and LC-QTOF-MS analysis under the conditions described below.

#### 2.2.2. LC-PDA

The chromatographic analysis was carried out using a methodology validated by Clever Leaves, a company dedicated to pharmaceutical grade cannabis-based products.

The liquid chromatography method with PDA (photodiode array) detection was employed, using the following conditions. Mobile phase A involved a solution of 0.1% trifluoroacetic acid in water, while mobile phase B involved a solution of acetonitrile. A total injection volume of 2 μL was used for the analysis. UV detection was set at a wavelength of 220 nm. Chromatographic separation was carried out on a CORTECS^®^ UPLC^®^ Shield RP18 column (Milford, USA) with dimensions of 2.1 × 100 mm and a particle size of 1.6 μm. The autosampler and column temperatures were maintained at 8 °C and 35 °C, respectively. The total run time for the analysis was 11 min. Acetonitrile HPLC was used as the solvent for dilutions, while a mixture of acetonitrile and water (70:30) was employed as solvent. The purge solvent consisted of a water–acetonitrile mixture (90:10). The flow rate was set at 0.7 mL/min, and the mobile phase composition was kept isocratic at 41% mobile phase A and 59% mobile phase B. The system suitability test required a resolution between peaks to be greater than 1.5 for proper analysis.

#### 2.2.3. Analysis by RP-LC-QTOF-MS

For metabolic analysis, 5 mg of the crude extract of *C. sativa*, which contains a high cannabidiol (CBD) content (>85% of the total phytocannabinoids extracted) [18], was dissolved in methanol to a final concentration of 250 mg/L for subsequent analysis via reverse-phase liquid chromatography coupled with mass spectrometry (RP-LC-QTOF-MS).

Samples were analyzed in a liquid chromatography system (Agilent Technologies 1260) coupled with a quadrupole time-of-flight (Q-TOF) mass analyzer (Agilent Technologies 6545B) with an electrospray ionization source (ESI). Separation was conducted in a C18 column (InfinityLab Poroshell 120 EC-C18 (100 × 3.0 mm, 2.7 µm) at 30 °C with a gradient elution consisting of 0.1% (*v*/*v*) formic acid in Milli-Q water (Phase A) and 0.1% (*v*/*v*) formic acid in acetonitrile (Phase B) at a constant flow rate of 0.4 mL/min. Mass spectrometric detection was performed initially in positive mode, followed by a subsequent analysis in negative mode using the same set of acquired data at full scan from 70 to 1100 *m*/*z*. The QTOF instrument was operated in 4 GHz (high resolution) mode. The data acquisition parameters were configured as follows: ion source temperature of 325 °C, gas flow of 8 L/min, nebulizer gas pressure at 50 psi, and capillary voltage of 2800 V. MS/MS acquisition mode was performed in data-dependent acquisition (DDA) mode in the range of *m*/*z* 50 to 1100 with a scan sweep rate of 3 spectra/s and under chromatographic and spectrometric conditions identical to those employed in the initial analysis. For each sample, analysis was performed at different collision energies 20 eV, 40 eV, and equation mode was used (CE = 3.6 × (*m*/*z*)/100 + 4.8) [59,60], using 3 precursors per cycle. During the analysis, several reference masses were used for mass correction: *m*/*z* 121.0509 (C_5_H_4_N_4_), *m*/*z* 922.0098 (C_18_H_18_O_6_N_3_P_3_F_24_) in positive mode and *m*/*z* 112.9856 [C_2_O_2_F_3_ (NH_4_)], and *m*/*z* 1033.9881 (C_18_H_18_O_6_N_3_P_3_F_24_) in negative mode.

#### 2.2.4. Data Processing

Data processing was performed with the Agilent MassHunter Profinder 10.0 software program for deconvolution, alignment, and integration, using the recursive feature extraction (RFE) algorithm. This algorithm performs a deconvolution of the chromatogram and integration of the molecular characteristics present in the samples according to mass and retention time. The data obtained from the deconvolution and integration were filtered by area by calculating the total area for the sample and then the area of each molecular feature. The annotation of the more abundant molecular features obtained was carried out using the CEU MASS MEDIATOR tool (https://ceumass.eps.uspceu.es/ (accessed on 1 October 2021)) [47], including the Metlin, Kegg, HDMB, and LipidMaps platforms as parameters, and with a tolerance of 10 ppm. Then, MS/MS analyses were performed in order to confirm the identity of the metabolites using MS-DIAL 4.8 (http://prime.psc.riken.jp/compms/msdial/main.html (accessed on 1 October 2021)), in in silico mass spectral fragmentation through CFM-ID 4.0 (https://cfmid.wishartlab.com/ (accessed on October 2021)) and manual MS/MS spectral interpretation using the Agilent MassHunter Qualitative Analysis program (version 10.0, USA).

#### 2.2.5. Cell Cytotoxicity and Anticancer Activity of *C. sativa* Leaf Extract

Cytotoxicity and anticancer activity were determined by analyzing the impact of *C. sativa* leaf extract on the metabolic activity of three different human carcinoma cell lines, namely gastric adenocarcinoma (AGS, ATCC^®^ CRL-1739), lung carcinoma (A549, ATCC^®^ CCL-185), and skin melanoma (A375, ATCC^®^ CRL-1619). Additionally, two healthy cell lines were employed, i.e., Vero (ATCC^®^ CCL-81) and human skin fibroblasts (HFF, ATCC^®^ SCRC-1041).

Cell viability was determined via a MTT metabolic activity assay (3-(4,5-Dimethylthiazol-2-yl)-2,5-Diphenyltetrazolium Bromide)) following the manufacturer’s instructions. For this, cells (7000–10,000 cells/well depending on the cell line) were seeded on 96-well microplates with supplemented culture medium (10% FBS) and then incubated at 37 °C, in a 5% CO_2_^,^ and humidified atmosphere (humidity above 90%) for 24 h. Next, the culture medium was extracted and replaced by a non-supplemented medium containing the *C. sativa* leaf extract at concentrations ranging from 0.05 to 0.0004 mg/mL (serial dilutions were performed). Cells were incubated at 37 °C, in a 5% CO_2_ and humidified atmosphere for 24 and 72 h. After the incubation time, 10 µL of the MTT reagent (5 mg/mL) was added to each well, and the microplates were incubated for 2 h under the same conditions. Finally, supernatants were extracted and replaced by 100 µL of DMSO to dissolve formazan crystals. Absorbance was recorded at 595 nm in a microplate reader (Multiskan™ FC Microplate Photometer, ThermoFisher Scientific, Waltham, MA, USA).

Cell viability was calculated using the following equation:$$Cellviability\left(\%\right)=100*\frac{Abs\left(C-\right)-Abs\left(sample\right)}{Abs\left(C-\right)}$$
where *Abs (C*−*)* corresponds to the absorbance of the negative control (non-supplemented medium) at 595 nm and *Abs (sample)* corresponds to the absorbance of the sample at 595 nm. In addition, *Cytotoxicity (%)* was calculated as 100 − *Cell viability (%*).

## 3. Results

### 3.1. Genome-Scale C. sativa Metabolic Reconstruction

The PlantSEED semi-automatic reconstruction strategy was performed and curated with an exhaustive review of the literature and BLASTKOALA, to obtain the first *C. sativa* GEM reported in the literature (Figure 1). Results were analyzed considering the challenges involved in modeling eukaryotic cells (large size, compartmentalization of metabolic processes, and variation in tissue-specific metabolic activity [61]) and also by considering topological characteristics of the network that can be analyzed from the stoichiometric matrix [62]. Features of the initially reconstructed network and topological analysis of the stoichiometric matrix through the sparsity pattern are shown in Table 1 and Table 2 and Figure 2.

Metabolic pathways with the highest number of reactions and compounds were associated with the biosynthesis of fatty acids, steroids, arginine, and tyrosine, along with the metabolism of purine, pyrimidine, glucose, starch, and sucrose (Figure 3).

#### 3.1.1. Functional Annotation

The initial genome annotation reported by Grassa contains 31,170 genes, of which 25,296 are protein-coding genes (81%). A PlantSEED functional annotation was performed and complemented via BLASTKOALA to describe the metabolic capacity of *C. sativa* leaves (Figure 4). A total of 10,636 *C. sativa* genes were related to KO numbers. Most orthologous groups were observed in metabolic pathways related to primary plant metabolism (amino acid, carbohydrate, energy, cofactors and vitamins, and lipid metabolism). *C. sativa* leaf metabolism reveals the complexity behind the biochemical reactions that occur in plant eukaryotic cells. A closer look at each of the modules (Figure 4) shows that energy acquisition, storage, and the utilization of stored energy are central processes in the overall control of plant metabolism [35]. Additionally, 6% of KO numbers were related to the biosynthesis of secondary cannabinoid and non-cannabinoid metabolites. These were important results that will be used to strengthen secondary metabolism in metabolic reconstruction.

Some of the metabolic modules manually added are biosynthesis of flavanone, flavonoids, tryptophane, catecholamine, phenylalanine, proline, arginine, valine, leucine, cholesterol, cannabinoids, and fatty acids, among others.

#### 3.1.2. Secondary Metabolites Biosynthesis of *C. sativa*

Figure 5 and Figure 6 allow the visualization of complex interactions involving different pathways in the metabolic network [56]. While terpenoids and cannabinoids share the metabolite geranyl pyrophosphate as a common precursor, coumarins and toxins originate from tryptophan and phenylalanine biosynthesis. Metabolic modules of phenylpropanoid biosynthesis, essential and non-essential amino acids such as tryptophan and tyrosine, biosynthesis of monoterpenes, terpenes, and sesquiterpenes could be responsible for the observed synergistic effects that enhance the bioactivities of cannabinoids (entourage effect).

After the reconstruction of the metabolic model, consecutive FBAs were conducted. The calculations are based on the constraints imposed by the stoichiometric coefficients of each reaction in the metabolic fluxes. The flux balance analysis approach is used to assess the ability of the model to predict the metabolic phenotypes of an organism under different conditions. A preliminary overview of FBA simulations of photosynthesis and photorespiration for the *C. sativa* model is evidenced in Appendix A. Thus far, 89.72% of the metabolic fluxes are active and 10.28% are blocked.

### 3.2. Non-Targeted LC-QTOF-MS Based Metabolomics Data

Characterization of the compounds present in the *C. sativa* leaf sample was performed using a non-targeted metabolomics approach. This approach has the advantage for the present study of analyzing the sample in general, without focusing on a particular set of metabolites, allowing for a more descriptive metabolomic characterization of the sample. Table 3 summarizes the identification of 41 molecules in negative ionization mode and 38 molecules in positive ionization mode. The metabolites obtained were clustered into four main clusters [64] of plant secondary metabolites (Figure 7)**.**

The molecules with the highest intensity in the abundance peaks were mostly cannabinoids (delta-9-THC, Cannabidiolic acid, Cannabichromene), and terpenoids (Geranylhydroquinone). However, high-intensity peaks were found for coumarins (clausarinol), phenylflavonoids (cannflavin A), and steroids (pregna-4,9(11)-diene-3,20-dione, Neriantogenin) (Table 3). Prenol lipids and glycerophospholipids were identified as the subgroups contributing to the greatest diversity of metabolites in the sample. The metabolic profile of the sample is illustrated in Appendix A. The main precursors in their biosynthesis were identified and integrated into the reconstruction (Table 4) and will be key to studying and understand the metabolic transition from primary to secondary metabolism and the relationship between chemical synergy and *C. sativa* valuable characteristics. Taking advantage of the reconstruction, it is possible to study the biosynthesis of various value-added compounds. From here, various approaches such as bio-organic synthesis can be used to obtain these valuable compounds in a more economical way.

### 3.3. Cytotoxicity and Anticancer Activity of C. sativa Leaf Extracts

The cytotoxicity of the *C. sativa* extract was clearly affected by different factors such as concentration, exposure time, and cell line. Results showed high anticancer activity against gastric adenocarcinoma (AGS) and melanoma cells (A375) (Figure 8).

Cytotoxicity levels ranging from 50 to 90% for concentrations between 0.0125 and 0.05 mg/mL were observed in both cell lines. In contrast, for Vero and lung carcinoma cells (A549), these cytotoxicity levels were observed in concentrations between 0.025 and 0.05 mg/mL. This confirms less activity against A549 and significant toxicity against Vero cells. Surprisingly, results obtained for healthy skin fibroblasts (HFF) showed negligible toxicity in concentrations between 0.0004 and 0.025 mg/mL (below 10%).

## 4. Discussion

### 4.1. GEM Reconstruction, Functional Annotation and Secondary Metabolism of C. sativa

The whole-genome assembly of *C. sativa* (CBDRx:18:580) obtained by Grassa et al. [44] serves as the main input for the GEM reconstruction. CBDRx:18:580 was obtained from the leaf of a female plant grown indoors at 20–25 °C [44]. The plant belongs to chemotype III, which is associated with a high content of cannabidiol-CBD [26]. While this plant chemotype is widely recognized for its applications in the textile and paper industry, there exist other significant avenues that present potential opportunities to diversify and enhance its value chain [6]. Some of these potential uses are in the food industry (thanks to its nutraceutical value), medicine (thanks to the unique properties of cannabinodiol), and cosmetics (thanks to the possible effects of cannabinoids in synergy with terpenes) [17].

As for the reconstruction, analysis of the corresponding stoichiometric matrix enables the identification of topological features of the network. The sparsity pattern is illustrated in Figure 2. The stoichiometric matrix consists of 1314 rows (metabolites) and 2101 columns (reactions). Out of a total of 2760714 entries, 8361 (0.302%) are non-zero (nz). Generally, fewer than 1% of the elements in a genome-scale stoichiometric matrix are non-zero. This value is particularly useful for comparing models based on the number of metabolites involved in each reaction. The double upper diagonal appearance observed in the stoichiometric matrix is primarily a result of the ordering of reactions, rather than an intrinsic feature [62,65]. GEM reconstruction of *C. sativa* incorporates various compartments, including the cytosol, stroma, Golgi, vacuole, cell wall, peroxisome, mitochondria, nucleus, and endoplasmic reticulum (Appendix A). Notably, around 50% of the model reactions specifically pertain to compartments that play crucial roles in primary metabolism, such as the cytosol, mitochondria, and plastids. Numerous studies have demonstrated the relationship between primary and secondary metabolisms in plants and how despite the great variety of secondary metabolites, only some basic pathways of primary metabolism function as their precursors [66]. Glycolysis is the precursor of fatty acid biosynthesis, the mevalonate pathway, and the DXP-MEP pathways, which give rise to a variety of important terpenes and phenolic compounds such as cannabinoids, flavonoids, and fatty acids. On the other hand, the Krebs cycle is a primary precursor in the biosynthesis of glutamate and aspartate, while the shikimate pathway is a precursor in the biosynthesis of phenylalanine, tyrosine, and tryptophan. This part of the relationship between primary and secondary metabolism is native to N-containing compounds. Thus, it is possible to affirm that glycolysis, Krebs cycle and shikimate pathways are the most important precursors in the reconstruction of the secondary metabolism of *C. sativa* when researching its potential as a cosmeceutical, cosmetic, or additive in the food industry (thanks to its properties derived from terpenes), therapeutic and nutraceutical (thanks to its properties derived from phenolic compounds such as flavonoids, cannabinoids, alkaloids and N-containing compounds), potential phytonutrient (thanks to its properties derived from fatty acids), and many other applications previously mentioned. A significant number of transport reactions were evidenced in the reconstruction, corresponding to the high flux of metabolites passing from one compartment to another. These reactions are linked to alkaloids, furanocoumarins, terpenes, and carotenoids formed in the chloroplast; similarly, sesquiterpenes, sterols, and hydroxylation steps together with fatty acid synthesis take place in a constant exchange between the cytosol and the endoplasmic reticulum. Most hydrophilic compounds originate in the cytosol, whereas the site of alkaloids, non-protein amino acids, glucosinolates, flavonoids, and carotenoids originate in the vacuole compartment [66].

### 4.2. Non-Targeted LC-QTOF-MS Based Metabolomics Data Analysis

In the present study, *C. sativa* chemotype III (with a phytocannabinoid content consisting of 12.78% CBD, 3.21% CBDA, and 0.54% THC as determined by LC-PDA) was chosen for the integration of metabolomics data into the reconstruction. The polar extract used in LC-QTOF-MS facilitates the identification of polar and low volatility compounds, mainly cannabinoids, some terpenes, and flavonoids [43]. In addition, the LC-QTOF-MS data show a complementarity between the positive and negative ionization modes. The positive ionization mode possibly reveals a higher quantity of CBD when compared to the quantity of THC. However, some THC isomers may play a role in peak discrimination. The metabolic profile obtained tentatively agrees with the initial chemotype III of the plant and also corresponds to the chemotype exposed by Grassa [44,67] in obtaining the reference genome of *C. sativa*.

LC-QTOF-MS data described high levels of secondary metabolites modules such as cannabinoids, terpenoids, coumarins, phenylpropanoids, and steroids and notable single metabolites such as delta-9-THC, cannabidiolic acid, cannabichromene, geranylhydroquinone, cannflavin A, pregna-4,9(11)-diene-3,20-dione, and neriantogenin.

The metabolites obtained from the analysis were found to exhibit a diverse range of chemical structures and functionalities. For the organization and classification of these metabolites, they were grouped into four distinct clusters. These clusters represent the main categories of plant secondary metabolites, highlighting the chemical diversity present in the sample [64] (Figure 7). This clustering approach provides valuable insights into the composition and distribution of secondary metabolites in the studied plant system [64].

#### 4.2.1. N-Containing Products

Approximately 24,000 known metabolites are considered part of the group of N-containing compounds [66,68]. These include alkaloids (21,000 known metabolites), amines, non-protein amino acids, cyanogenic glycosides, glucosinolates, alkamides, lectins, peptides, and polypeptides.

Alkaloids can be defined as nitrogen-containing compounds derived from secondary, or specialized, metabolism. Their nitrogen compound is derived from an amino acid, and they are part of a complex ring structure [69]. Although there is an immense diversity of alkaloids, they all share a biosynthetic origin, derived from the formation and reactivity of the iminium cation. Its transition from primary to secondary metabolism is considered the most important as it opens the door to a new chemical space [54]. The first of the four stages found in alkaloid biosynthesis consists of the accumulation of an amino precursor from amino acid metabolism; these amino precursors can be divided into two categories: polyamines derived from lysine, arginine, and ornithine or aromatic amines derived from tryptophan and tyrosine. In the first case, polyamides are produced through the Krebs cycle pathway, which generates aspartate as a precursor of lysine and pyrimidines, as well as glutamate, which functions as a precursor of ornithine, arginine, and non-protein amino acids [66]. In the second case, the aromatic amines come from the shikimate pathway, which produces chorismate as a precursor, on one hand from arogenate to produce tyrosine and phenylalanine and on the other hand from anthranilate to produce tryptophan [66] (Figure 6). Tyrosine is the precursor to multiple alkaloid families, including the benzylisoquinolines, the amaryllidaceae alkaloids, and the betalains.

LC-QTOF-MS data revealed that neurine and cannabisativine are two alkaloids present in the leaves of the plant, in addition to their previously reported presence in the root of samples collected in Mexico [12,70]. These cannabis alkaloids have demonstrated antiparasitic, antipyretic, antiemetic, antitumor, diuretic, and analgesic properties [11,71]. Neurine can be biosynthesized from choline [72], which has been classified as an essential nutrient for humans, and additionally, it is a precursor of the osmoprotectant glycine betaine, an enhancer of osmotic resistance in the plant against drought and salinity [73]. Choline biosynthesis is thus a potential nutraceutical pathway by which 3 methylation reactions occur, catalyzed in parallel by the cytosolic enzyme phosphoethanolamine N-methyltransferase (EC 2.1.1.103) and mediating the next 2 methylations to produce phosphocholine [73]. Other N-containing compounds obtained in the LC-QTOF-MS data were glyceryl lactopalmitate, which is used in the food industry as an emulsifier [74] and belongs to the pyrazole-type alkaloids from ornithine. Another compound identified was pipercitine, which has proven insecticidal activity [75] and can be obtained from lysine.

In plants, between 20 and 30% of fixed carbon is invested in the synthesis of phenylalanine and then converted into lignin, which fulfills different roles in structural function as the most abundant compound in the cell wall, ultraviolet protection, signaling, and reproduction thanks to volatile anthocyanins and phenylpropanoid/benzenoid [76]. The latter is the second largest group of volatiles in plants, and they are divided into three classes according to their carbon backbone: benzenoids (C6–C1), phenylpropanoids (C6–C3), and phenylpropanoid-related compounds (C6–C2) [77]. Their biosynthesis is based on the amino acid-derivative pathways of shikimic acid (E.C. 1.1.1.25), which consists of seven reactions catalyzed by six enzymes and transforms phosphoenolpyruvate (PEP) with erythrose 4-phosphate (E4P) to chorismite (Figure 6).

#### 4.2.2. Phenolic Compounds: Polyphenols, Phenylpropanoids, Flavonoids

More than 10,000 different structures related to phenolic compounds have been identified [7]. Forty-two phenolic compounds have been identified in *C. sativa* [78], of which twenty-six different flavonoids have been identified [8,16], belonging mainly to two classes, favonols and favones [79]. The seven chemical structures of the flavonoid aglycones are orientin, vitexin, isovitexin, apigenin, luteolin, kaempferol, and quercetin.

These phenolic compounds share precursors with compounds derived from the nitrogenous pathways and include enzymes such as phenylalanine-ammonia-lyase (PAL), cinnamate 4-hydroxylase (C4H, a cytochrome P450) and 4-coumarate-CoA ligase (4CL) (Figure 6). These enzymes transform the aromatic amino acids phenylalanine and tyrosine into coenzyme A-activated 4-coumaric acid via the phenyl-propanoid pathway [53]. 4-Coumaroyl-CoA gives rise to many different natural products. These include flavones, aglycones in the form of O- and C-glycosides such as apigenin-8-C-glucoside [80], cannflavin A produced by enzymatic precursors such as caffeoyl CoA and feruloyl CoA, and ligands such as secoisolariciresinol, cannabisin D.

Data extracted from LC-QTOF-MS described some of the unique metabolites of the species, such as cannflavin A (Table 3). Cannaflavins come from the condensation of three malonyl molecules to form naringenin chalcone. When the ring is closed, it forms naringenin and thanks to the action of flavone synthase, it is possible to produce apigenin, which is a derivative of luteonyl [16]. Among the reported benefits of flavonoids and particularly cannabiflavins are their antioxidant and anti-inflammatory activity, and cardioprotective, neuroprotective, hepatoprotective, and immunomodulatory effects [80]. Other properties of flavonoids are their flavor, color, and aroma, as well as anti-diabetic and neuroprotective activities thanks to the modulation of the number of cellular cascade signals [11].

However, there are gaps in our knowledge of the biosynthesis of flavonoids and therefore the means by which some esters, lignins, flavonoids, and coumarins are formed is unknown [7].

#### 4.2.3. Fatty Acids Derivates

Fatty acids are often esterified in form of phospholipids, glycerolipids, or sterol backbones. Their structure consists of a long chain of hydrogen-bonded carbons, with a terminal carboxyl group (-COOH) [11]. This functional group is key in their function as energy reservoirs. In this regard, they provide structure to and energy for cells in the absence of glucose and participate in the response to low-temperature tolerance. Finally, they are involved in the production of cholesterol as precursor for the biosynthesis of hormones such as estrogen, testosterone, vitamin D hormone, steroids, and prostaglandins [55]. These functions also explain their high nutritional value and pharmaceutical potential.

Fatty acids are synthesized in plastids and assembled by glycerolipids or triacylglycerols in the endoplasmic reticulum [81]. Fatty acid synthesis is a complex process involving three main phases: de novo synthesis of fatty acids in the plastidial compartment from acetyl CoA, desaturation in the chloroplast and elongases, modified reactions such as hydroxylation, and epoxidation, which take place in the endoplasmic reticulum [82]. Figure 5 and Figure 6 describe in general terms the metabolism of fatty acid biosynthesis.

About 22% of metabolites detected in this non-targeted LC-QTOF-MS metabolomic analysis of a *C. sativa* sample are involved in different reactions related to the fatty acid biosynthesis. As products of de novo fatty acid synthesis, palmitoleic acid and other linolenic acids (13-Hydroxyoctadecatrienic acid, octadecatetraenoic acid, and trihydroxy-octadecadienoic acid) were identified. It has been reported that increasing the dietary intake of these fatty acids reduces the risk of coronary heart disease [83] due to inhibition of coagulation, improvement of glucose homeostasis, and attenuation of inflammation. On the other hand, fatty acids metabolized via modifiable reactions increase the production of vitamin E, prostacyclin, prostaglandins, leukotrienes, and hydroxy and hydroperoxy fatty acids, which have been reported to be involved in the modulation of cell growth, angiogenesis, inflammation, thrombosis, immune response, inhibition of carcinogenesis and tumor growth, and stimulation of cancer cells apoptosis, among others [84,85].

#### 4.2.4. Terpenes

Terpenes are hydrocarbon compounds made up of 5C units called isoprenes. They are classified according to these units’ size. Their biosynthesis is mediated by the cytosolic mevalonate (MVA) pathway, which provides farnesyl diphosphate (FPP) for sesquiterpenoids (C15) and squalene as precursors for triterpenoids (C30) and sterols. Alternatively, they might come from the patricidal DOXP/MEP pathway, which provides GPP to form monoterpenoids [8] (Figure 6). Almost 30% of the data obtained via LC-QTOF-MS are related to various terpenes and terpenoids. These compounds have shown multiple therapeutic benefits, including suppressing the immune system response against COVID-19, and inhibition in many species of bacteria and fungi [11]. Additionally, they have been reported to exhibit antimicrobial, repellant, antiallergy, anticancer, antifungal, antibacterial, antioxidant, anti-inflammatory, antidepressant, sedative, anticonvulsant, analgesic, gastroprotective, and antispasmoic properties [11].

The main precursors of the metabolites identified from LC-QTOF-MS metabolomics data were described and used for integration in the metabolic reconstruction (Table 4). The integrated data are mainly primary precursors for metabolic modules of interest: anthocyanin biosynthetic pathway which is an extension of flavonoid pathway; fatty acid biosynthesis, degradation, and elongation; phenylalanine, tyrosine, and tryptophan biosynthesis; and terpenoid backbone biosynthesis. After data integration, the reconstruction increased by 297 active reactions and 118 metabolites.

### 4.3. Cytotoxicity and Anticancer Activity of C. sativa Leaf Extracts

The obtained results confirmed the remarkable anticancer activity of the *C. Sativa* extracts against different carcinoma cell lines (AGS, A375 and A549). This agreed well with previous works that studied the anticancer activity of *C. sativa* on different cell lines such as melanoma [20], ovarian cancer [21], prostate cancer [22], and breast and pancreatic cancer [16], among others. The results are also in agreement with the biological activities based on both the chemotype and the extraction taken from the leaves of the plant. Manosroi et al. [86] demonstrated that the ethanolic extract of the leaves and seeds of the *C. sativa* plant chemotype III, exhibited cytotoxicity activity against B16F10 melanoma cells in a concentration dependent manner (cytotoxicity of 46% at 1 mg/mL and total inhibition at 10 mg/mL). Additionally, both leaf and seed extracts demonstrated negligible toxicity against human skin fibroblast (viability above 80% for concentration below 0.5 mg/mL) confirming high biocompatibility.

The notable activity against melanoma cells combined with the negligible impact on healthy human skin cells confirms the great pharmacological potential that makes them suitable candidates for the development of new-generation topical treatments with reduced side effects, especially for melanoma, the most common and aggressive type of skin cancer. These findings have been confirmed in several works presenting promising results, both in vitro [87] and in vivo [20].

On the other hand, the potential selective toxicity of *C. sativa* leaf extracts has been widely studied in order to develop novel therapies with reduced negative side effects. Janatová and colleagues [15] evaluated selectivity by comparing the toxicity of six different genotypes of medical cannabis against three cancer cell lines (Ht-29, Caco-2, and Hep-G2) and two healthy cell lines (FHs 74 Int: healthy intestinal cells and MRC-5: healthy lung fibroblast). They demonstrated that the compound content of the different genotypes strongly affects selectivity. Highlighting specific compounds such as myrcene, β-elemene, β-selinene, and α-bisabolol oxid as enhancers of selectivity and β-ocimene and β-caryophyllene oxide as cytotoxicity-associated molecules. Selectivity is therefore determined by the plant genotype (chemical profile and content) and by the specific cell line.

In consequence, these findings can explain the selectivity differences between all the different evaluated cell lines, especially, the significant increase of cytotoxicity observed in Vero cells. Furthermore, the obtained toxicity profiles against Vero cells agree strongly with previously reported articles. For example, Lamdabsri and coworkers [88] showed that the toxicity of cannabis extracts against Vero cells is highly influenced by compound content, reporting high toxicity in the crude and CBN extracts (IC50 of 13.4 and 10.6 μg/mL, respectively) and lower toxicity in the CBG, CBD, and THC (IC50 699.7, 39.77 and 67.2 μg/mL, respectively).

## 5. Conclusions

GEM reconstruction of *C. sativa* contributes to better understanding of cellular phenotypes and metabolic behavior [41,89] in terms of the identification of different biosynthetic pathways by integrating omics data and experimental anticancer results. Using the current model, it is possible to explore different biosynthetic pathways for many valuable compounds, especially those of major interest to the scientific community and which represent a significant opportunity to improve the value chain for *C. sativa*. The high number of reactions observed in the cytosol, plastids, and mitochondria compartments confirms the significance of primary metabolic pathways such as glycolysis, the Krebs cycle, and the shikimate pathway. These pathways play a crucial role as principal precursors for secondary metabolites, including cannabinoids, flavonoids, fatty acids, and nitrogen-containing compounds. Transport reactions have a crucial role in facilitating the exchange of metabolites between different cellular compartments. This is especially important in compartments such as the chloroplast, cytosol, endoplasmic reticulum, and vacuole, which are related to the synthesis of various metabolites, including alkaloids, terpenes, sterols, and hydrophilic compounds.

On the other hand, the LC-QTOF-MS metabolomics analysis provided insights into the diverse chemical composition and distribution of secondary metabolites in *C. sativa.* The LC-QTOF-MS data revealed a high abundance of secondary metabolite modules such as cannabinoids, terpenoids, coumarins, phenylpropanoids, and steroids. Specific metabolites identified included delta-9-THC, cannabidiolic acid, cannabichromene, geranylhydroquinone, cannflavin A, pregna-4,9(11)-diene-3,20-dione, and neriantogenin. These metabolites exhibit a range of biological activities and potential therapeutic benefits. Additionally, these metabolites contributed to the integration of the reconstruction, demonstrating that the use of omics contributes to the activation of a greater number of reactions that are required for the synthesis of metabolites in the reconstruction.

Finally, regarding to the cytotoxicity and anticancer activity of *C. sativa*, it can be concluded that although extracts demonstrated low selectivity in Vero cells, their remarkable selectivity against melanoma cells compared to the healthy skin fibroblast leaves an open window for continuing studies on *C. sativa* leaf extract as a potential candidate for the development of new-generation treatments for skin cancer with reduced side effects.

## Figures and Tables

**Figure 1 metabolites-13-00788-f001:**
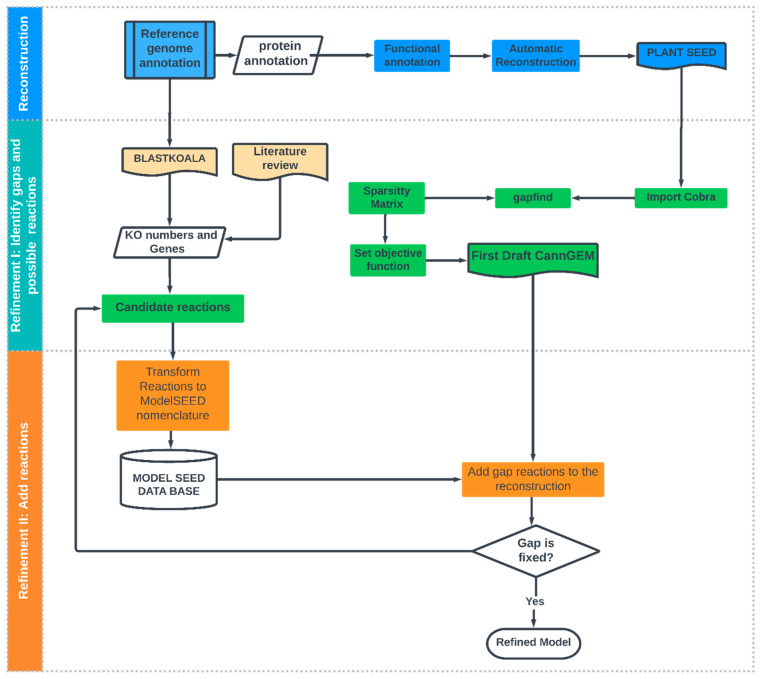
Iterative workflow from the bottom-up process in *C. sativa* reconstruction.

**Figure 2 metabolites-13-00788-f002:**
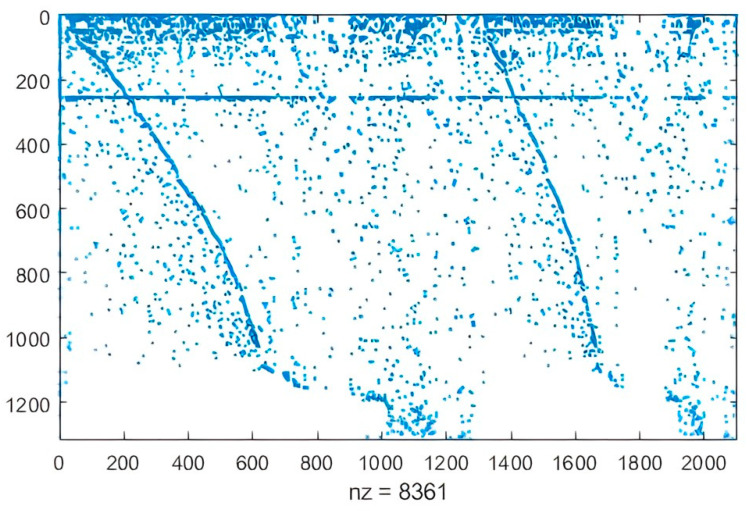
Stoichiometric matrices of the implemented model. The y axis represents the number of metabolites, and the x axis represents the chemical reactions.

**Figure 3 metabolites-13-00788-f003:**
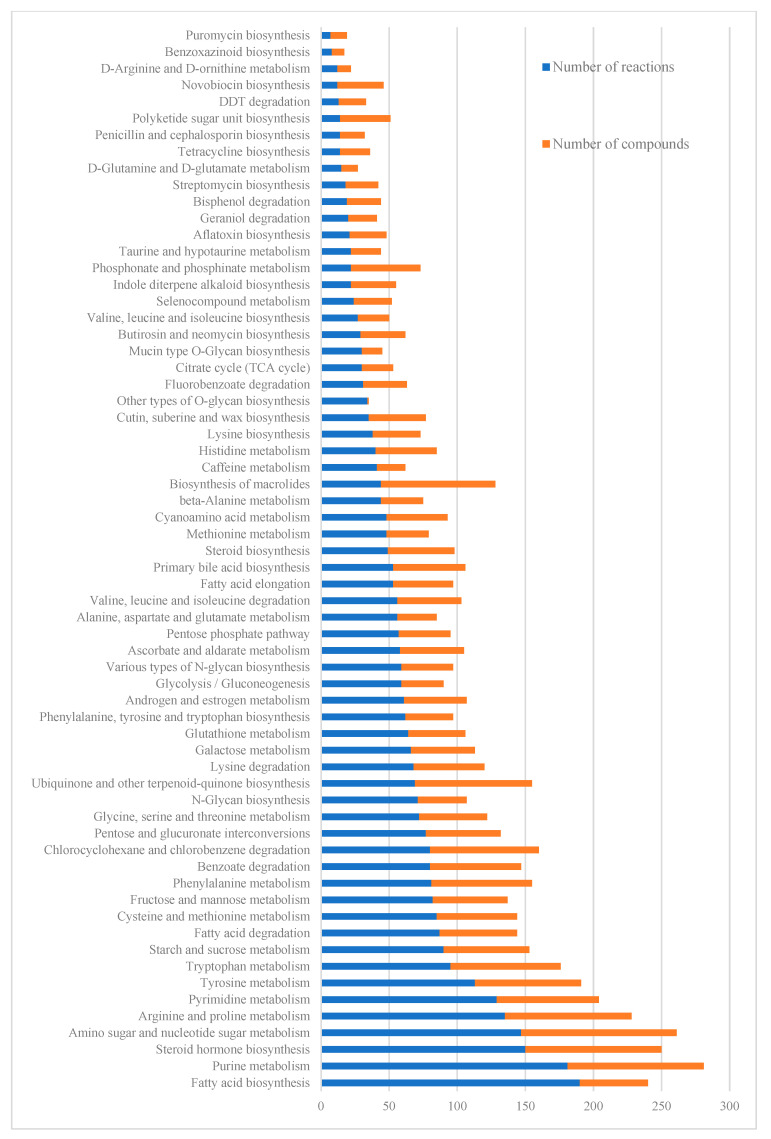
Reactions and compounds of the *C. sativa* metabolic reconstruction grouped by pathways according to PlantSEED.

**Figure 4 metabolites-13-00788-f004:**
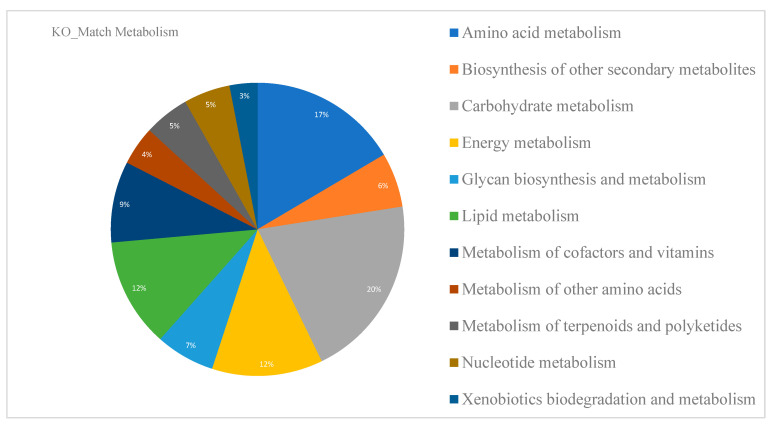
Molecular functions stored in the KO (KEGG Orthology) database containing orthologs of experimentally characterized genes/proteins in the reference genome of *Cannabis sativa*.

**Figure 5 metabolites-13-00788-f005:**
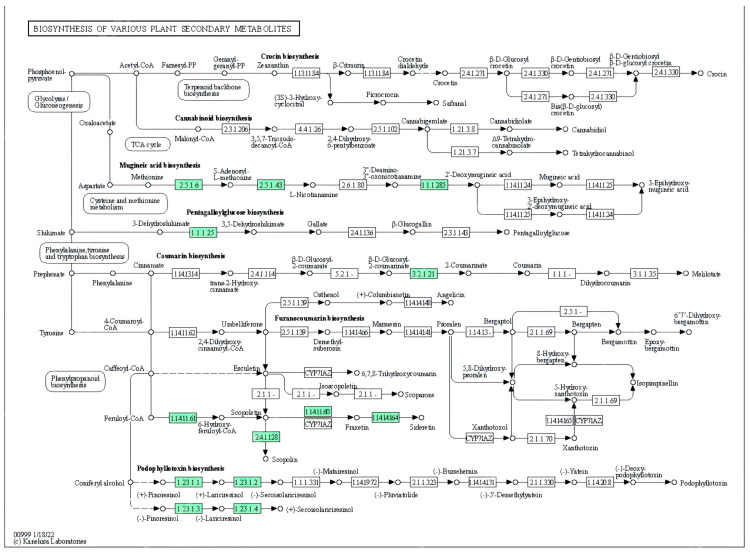
Pathway of secondary metabolism in the biosynthesis of cannabinoids and terpenes in *C. sativa* [63]. Enzymes related to functional annotation are illustrated in green.

**Figure 6 metabolites-13-00788-f006:**
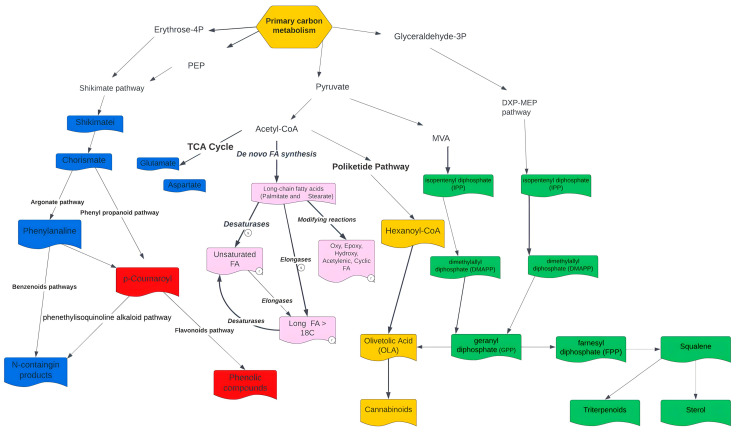
Secondary metabolism of *C. sativa,* classified by five major metabolite types: terpenes (green), fatty acids (pink), phenolic compounds (red), N-compounds (blue), and cannabinoids (yellow).

**Figure 7 metabolites-13-00788-f007:**
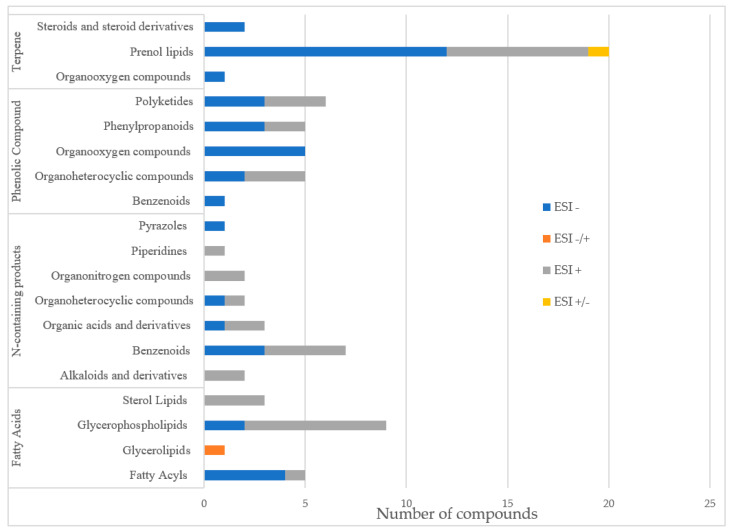
Bar diagram with the number of compounds identified in *C. sativa* leaf extract obtained from legal cultivation from Clever Leaves, via LC–QTOF MS/MS. In blue: electrospray negative ionization mode; in orange: electrospray switching polarity mode; in gray, electrospray positive ionization mode; in yellow, electrospray switching polarity mode.

**Figure 8 metabolites-13-00788-f008:**
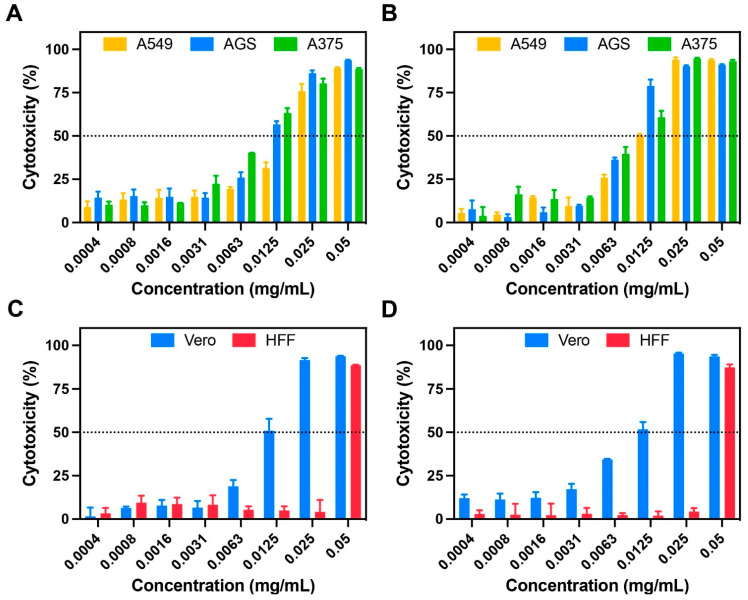
Cytotoxicity analysis of the *C. sativa* leaf extract on different carcinoma and healthy cell lines. Results for carcinoma cell lines (A549, AGS and A375) after 24 (**A**) and 72 h of exposure (**B**). Results for healthy cell lines (Vero and HFF) after 24 (**C**) and 72 h of exposure (**D**).

**Table 1 metabolites-13-00788-t001:** Network characteristics of the reconstructed metabolic network: Comparison among *C. sativa* metabolic reconstruction and AraGEM. c cytosol, d stroma, g Golgi, v vacuole, w cell wall, x peroxisome, m mitochondria, n nucleus, r endoplasmic reticule, u unknown, GPR gene-protein reaction. Appendix A.

	Strategy—Plant Seed	AraGEM
Reactions	2101	1567
Metabolites	1314	1748
GPR	1462	5253
Transport Reactions	143	148
Compartments	c, d, g, v, w, x, m, n, r, e, j	c, m, p, x, plastid, v

**Table 2 metabolites-13-00788-t002:** Curation statistics of the models, using gapfind algorithm with Cobra Toolbox [47].

	Strategy II—Plant Seed
allGaps	228
rootGaps	100
downstreamGaps	128

**Table 3 metabolites-13-00788-t003:** Compounds identification of LC-MS metabolomic data from a *C. sativa* leaf sample cultivated in a licit operation in Colombia. Analytical platform LC-QTOF-MS.

Compound	Formula	Mass	RT (min)	Mass Error (ppm)	Adduct	DET	ID Confidence ^a^	Area (%) ^b^
*Alkaloids and derivatives*								
Neurine	C_5_H_13_NO	103.0997	1.09	7	[M+H]^+^	ESI +	Level 3	0.86
Cannabisativine	C_21_H_39_N_3_O_3_	381.2991	5.66	4	[M+H]^+^	ESI +	Level 3	0.31
*Benzenoids*								
Phenylacetaldehyde	C_8_H_8_O	120.0575	3.33	5	[M+H-H_2_O]^+^	ESI +	Level 3	0.43
Methylstyrene	C_9_H_10_	118.0783	14.59	6	[M+H]^+^	ESI +	Level 3	2.31
Phenylpropanal	C_9_H_10_O	134.0732	14.59	5	[M+H]^+^	ESI +	Level 3	0.65
Cyclointegrin	C_21_H_20_O_6_	368.1260	14.87	0	[M-H]^−^	ESI −	Level 3	0.61
Cresol	C_7_H_8_O	108.0575	16.46	4	[M-H]^−^	ESI −	Level 3	1.25
Levomethadyl Acetate	C_23_H_31_NO_2_	353.2355	16.99	6	[M+H]^+^	ESI +	Level 3	1.17
Hydroxy-(pentadecatrienyl)benzoic acid	C_22_H_30_O_3_	342.2195	17.33	4	[M-H]^−^	ESI −	Level 3	0.30
*Fatty Acyls*								
Corchoionol C glucoside	C_19_H_30_O_8_	386.1941	6.17	0	[M-H]^−^	ESI −	Level 2	0.46
Trihydroxy-octadecadienoic acid	C_18_H_32_O_5_	328.2250	11.14	1	[M-H]^−^	ESI −	Level 2	0.42
Octadecatetraenoic acid	C_18_H_28_O_2_	276.2089	15.23	1	[M-H]^−^	ESI −	Level 3	0.37
Hydroxyoctadecatrienic acid	C_18_H_30_O_3_	294.2195	15.32	1	[M-H-H_2_O]^−^	ESI −	Level 3	1.40
Palmitoleic acid	C_16_H_30_O_2_	254.2246	16.89	5	[M+H-H_2_O]^+^	ESI +	Level 3	0.79
*Glycerolipids*								
Gingerglycolipid A	C_33_H_56_O_14_	676.3670	14.50	1	[M-H]^−^	ESI −/+	Level 3	0.72
*Glycerophospholipids*								
LPC 16:0	C_24_H_50_NO_7_P	495.3325	15.94	6	[M+H]^+^	ESI +	Level 2	0.43
LPC 8:0	C_16_H_32_NO_8_P	397.1866	16.21	2	[M+HCOOH-H]^−^	ESI −	Level 3	0.98
PI 41:7	C_50_H_83_O_13_P	922.5571	16.03	7	[M+HCOOH-H]^−^	ESI −	Level 3	0.39
PS O-37:2	C_43_H_82_NO_9_P	787.5727	16.51	2	[M+Na]^+^	ESI +	Level 3	0.39
PE 38:5	C_43_H_76_NO_8_P	765.5309	16.56	3	[M+H]^+^	ESI +	Level 3	0.49
PA O-36:4	C_39_H_71_O_7_P	682.4937	16.59	4	[M+Na]^+^	ESI +	Level 3	0.72
PA O-36:6	C_39_H_67_O_7_P	678.4624	16.64	6	[M+H-H_2_O]^+^	ESI +	Level 3	0.78
LPG 16:0	C_22_H_45_O_9_P	484.2801	16.96	10	[M+H]^+^	ESI +	Level 3	0.62
PG 25:3;O3	C_31_H_55_O_13_P	666.3380	16.67	9	[M+H]^+^	ESI +	Level 3	0.37
*Organic acids and derivatives*								
Alloisoleucine	C_6_H_13_NO_2_	131.0946	1.89	3	[M-H]^−^	ESI −	Level 3	0.35
Dilauryl 3.3′-thiodipropionate	C_30_H_58_O_4_S_2_	546.3777	16.74	3	[M+H-H_2_O]^+^	ESI +	Level 3	0.60
Gly-Tyr-Tyr-Pro-Thr	C_29_H_38_N_5_O_9_	600.2670	16.99	6	[M+Na]^+^	ESI +	Level 3	0.61
*Organoheterocyclic compounds*								
delta-9-THC	C_21_H_30_O_2_	314.2246	15.76	5	[M+H]^+^	ESI +	Level 2	3.71
delta-9-THC	C_21_H_30_O_2_	314.2246	16.47	5	[M+H]^+^	ESI +	Level 2	5.67
Geranylhydroquinone	C_16_H_22_O_2_	246.1620	16.46	0	[M-H-H_2_O]^−^	ESI −	Level 3	13.35
methyl-(4-methylpent-3-en-1-yl)-2H-chromen-ol	C_16_H_20_O_2_	244.1463	16.46	2	[M-H]^-^	ESI −	Level 3	3.10
Dimethyl-prenylchromene -carboxylic acid	C_17_H_20_O_3_	272.1413	16.46	2	[M-H]^−^	ESI −	Level 2	2.09
Phaeophorbide b	C_35_H_34_N_4_O_6_	606.2478	17.19	4	[M+H]^+^	ESI +	Level 3	0.55
*Organonitrogen compounds*								
Tetradecylamine	C_14_H_31_N	213.2457	14.11	6	[M+H]^+^	ESI +	Level 3	0.79
Palmitoleoyl-EA	C_18_H_35_NO_2_	297.2668	16.31	8	[M+Na]^+^	ESI +	Level 3	0.34
*Organooxygen compounds*								
Trehalose	C_12_H_22_O_11_	342.1162	1.13	0	[M-H]^−^	ESI −	Level 2	0.93
Kobusone	C_14_H_22_O_2_	222.1620	15.83	3	[M-H]^−^	ESI −	Level 2	0.51
Methyl-pentenone	C_6_H_10_O	98.0732	16.46	1	[M-H-H_2_O]^−^	ESI −	Level 3	0.75
Methylpicraquassioside A	C_19_H_24_O_10_	412.1369	16.61	10	[M+Cl]^−^	ESI −	Level 3	0.68
(carboxymethoxy)- trihydroxyoxane-carboxylic acid	C_8_H_12_O_9_	252.0481	16.81	1	[M+HCOOH-H]^−^	ESI −	Level 3	0.61
Epoxyprogesterone	C_21_H_28_O_3_	328.2038	17.39	2	[M-H]^−^	ESI −	Level 3	0.59
*Phenylpropanoids*								
Clausarinol	C_24_H_30_O_6_	414.2042	14.59	4	[M+H]^+^	ESI +	Level 3	4.45
6-{[2-(dihydroxyphenyl)-3-(dimethylocta-dien-yl)-hydroxy-(3-methylbut-2-en-yl)-4-oxo-4H-chromen-6-yl]oxy}-trihydroxyoxane-carboxylic acid	C_36_H_42_O_12_	666.2676	15.58	1	[M-H]^−^	ESI −	Level 3	0.33
Nevskin	C_24_H_32_O_5_	400.2250	16.27	1	[M+H]^+^	ESI +	Level 3	0.60
Methoxy-abietatrienolide	C_21_H_28_O_3_	328.2038	16.48	1	[M-H]^−^	ESI −	Level 3	0.47
Nordihydroguaiaretic acid	C_18_H_22_O_4_	302.1518	16.65	2	[M-H]^−^	ESI −	Level 3	0.32
*Piperidines*								
Pipercitine	C_23_H_43_NO	349.3345	15.91	5	[M+H-H_2_O]^+^	ESI +	Level 3	4.88
*Polyketides*								
Cannabidiolic acid	C_22_H_30_O_4_	358.2144	16.26	4	[M+H-H_2_O]^+^	ESI +	Level 3	7.15
Cannflavin A	C_26_H_28_O_6_	436.1886	16.34	4	[M+H]^+^	ESI +	Level 3	1.84
Betavulgarin	C_17_H_12_O_6_	312.0634	16.34	3	[M+H]^+^	ESI +	Level 3	1.18
Cannflavin A	C_26_H_28_O_6_	436.1886	16.41	4	[M+H]^+^	ESI −	Level 3	2.55
Chlorophorin	C_24_H_28_O_4_	380.1988	16.46	8	[M+HCOOH-H]^−^	ESI −	Level 3	0.53
Quercetol B	C_23_H_28_O_4_	368.1988	16.51	3	[M-H]^−^	ESI −	Level 3	0.67
*Prenol lipids*								
Icariside B8	C_19_H_32_O_8_	388.2097	6.19	2	[M-H]^−^	ESI −	Level 3	0.34
Capsularone	C_27_H_38_O_8_	490.2567	11.77	1	[M+HCOOH-H]^−^	ESI −	Level 3	0.66
Diterpenoid EF-D	C_27_H_38_O_7_	474.2618	13.60	1	[M+HCOOH-H]^−^	ESI −	Level 3	1.13
Persicachrome	C_25_H_36_O_3_	384.2664	14.33	4	[M+H-H_2_O]^+^	ESI +	Level 3	0.72
Nigellic acid	C_15_H_2_0O_5_	280.1311	14.59	3	[M+H]^+^	ESI +	Level 3	0.37
Yucalexin	C_20_H_26_O_4_	330.1831	15.67	4	[M-H]^−^	ESI −	Level 3	0.49
2-(Hydroxy-methylphenyl)-5-methyl-4-hexen-3-one	C_14_H_18_O_2_	218.1307	15.67	7	[M-H]^−^	ESI −	Level 3	0.38
Tintinnadiol	C_21_H_32_O_3_	332.2351	15.76	1	[M-H]^−^	ESI −	Level 3	1.21
Hydroxymethylphenyl pentanone	C_12_H_16_O_2_	192.1150	15.76	5	[M+H]^+^	ESI +	Level 2	0.32
Dimethylrosmanol	C_22_H_30_O_5_	374.2093	16.00	1	[M-H]^−^	ESI −	Level 2	0.64
hydroxy-methoxy-(3-methylbut-2-en-1-yl)benzoic acid	C_13_H_16_O_4_	236.1049	16.26	4	[M+H-H_2_O]^+^	ESI +	Level 2	0.77
Lucidone B	C24H32O5	400.2250	16.28	1	[M-H]^−^	ESI −	Level 2	1.53
Pentylresorcinol	C_11_H_16_O_2_	180.1150	16.46	3	[M-H]^−^	ESI −	Level 2	2.06
Hyperforin	C_35_H_52_O_4_	536.3866	16.46	1	[M-H-H_2_O]^−^	ESI −	Level 3	0.30
Hydroxymethylphenyl)pentanone	C_12_H_16_O_2_	192.1150	16.47	5	[M+H]^+^	ESI +/-	Level 2	0.76
Curzerenone	C_15_H_18_O_2_	230.1307	16.48	3	[M-H]^−^	ESI −	Level 3	0.54
Geranyl benzoate	C_17_H_22_O_2_	258.1620	16.49	6	[M+H]^+^	ESI +	Level 3	0.33
Hydroxy- Caroten-3′-one	C_40_H_54_O	550.4175	16.71	9	[M+Na]^+^	ESI +	Level 3	0.73
Trimethyl-pentadecatrien-2-one	C_18_H_30_O	262.2297	17.16	6	[M+H]^+^	ESI +	Level 2	0.82
Grifolin	C_22_H_32_O_2_	328.2402	17.66	3	[M-H]^−^	ESI −	Level 3	0.32
*Pyrazoles*								
Glyceryl lactopalmitate	C_20_H_16_N_6_O_2_S	404.1055	16.23	8	[M+HCOOH-H]^−^	ESI −	Level 3	1.38
*Steroids and steroid derivatives*								
Pregnadienedione	C_21_H_28_O_2_	312.2089	16.48	0	[M-H]^−^	ESI −	Level 3	2.24
Neriantogenin	C_23_H_32_O_4_	372.2301	17.66	2	[M-H]^−^	ESI −	Level 3	2.89
*Sterol Lipids*								
Rhodexin A	C_29_H_44_O_9_	536.2985	15.43	1	[M+H]^+^	ESI +	Level 3	0.45
ST 27:0;O7	C_27_H_48_O_7_	484.3400	16.77	4	[M+H]^+^	ESI +	Level 3	0.35
Dihomocholic acid	C_26_H_44_O_5_	436.3189	17.30	8	[M+Na]^+^	ESI +	Level 3	0.53

RT: retention time; LC: liquid chromatography; QTOF-MS: quadrupole time-of-flight mass spectrometer. ^a^: Identification confidence levels: Level 1: Confirmed structure, Level 2: Probable structure, Level 3: Tentative candidates(s), Level 4: Unequivocal molecular formula, Level 5: Exact mass. ^b^: The data obtained from the deconvolution and integration were filtered by area by calculating, total area for the sample and then area of each molecular feature.

**Table 4 metabolites-13-00788-t004:** Validation of the metabolic reconstruction using precursors of metabolites detected in LC-QTOF-MS data.

	Precursor	EC Number	Is It Included in CannGEM?
Anthocyanin biosynthesis	BZ1; anthocyanidin 3-O-glucosyltransferase	2.4.1.115	No
	3MaT1; anthocyanin 3-O-glucoside-6″-O-malonyltransferase	2.3.1.171	No
	3MaT2; anthocyanidin 3-O-glucoside-3″,6″-O-dimalonyltransferase	2.3.1.-	No
	3GGT; anthocyanidin 3-O-glucoside 2″-O-glucosyltransferase	2.4.1.297	No
	5GT; cyanidin 3-O-rutinoside 5-O-glucosyltransferase	2.4.1.116	No
	AA7GT; cyanidin 3-O-glucoside 7-O-glucosyltransferase (acyl-glucose)	2.4.1.300	No
	UGT79B1; anthocyanidin 3-O-glucoside 2′″-O-xylosyltransferase	2.4.2.51	No
	3AT; anthocyanidin 3-O-glucoside 6″-O-acyltransferase	2.3.1.215	No
	5MaT1; anthocyanin 5-O-glucoside-6′″-O-malonyltransferase	2.3.1.172	No
	5MaT2; anthocyanin 5-O-glucoside-4′″-O-malonyltransferase	2.3.1.214	No
	UGT75C1; anthocyanidin 3-O-glucoside 5-O-glucosyltransferase	2.4.1.298	No
	AA5GT; cyanidin 3-O-glucoside 5-O-glucosyltransferase (acyl-glucose)	2.4.1.299	No
	5AT; anthocyanin 5-aromatic acyltransferase	2.3.1.153	No
	UGAT; cyanidin-3-O-glucoside 2″-O-glucuronosyltransferase	2.4.1.254	No
	GT1; anthocyanidin 5,3-O-glucosyltransferase	2.4.1.-	Yes
	3GT; anthocyanin 3′-O-beta-glucosyltransferase	2.4.1.238	No
Fatty acid biosynthesis	ACACA; acetyl-CoA carboxylase	6.4.1.2	Yes
	ACSF3; malonyl-CoA/methylmalonyl-CoA synthetase	6.2.1.-	Yes
	FASN; fatty acid synthase, animal type	2.3.1.85	Yes
	FAS1; fatty acid synthase subunit beta, fungi type	2.3.1.86	Yes
	fas; fatty acid synthase, bacteria type	2.3.1.-	No
	HT2; 3-hydroxyacyl-thioester dehydratase, animal type	4.2.1.-	No
	FATB; fatty acyl-ACP thioesterase B	3.1.2.14	Yes
	FATA; fatty acyl-ACP thioesterase A	3.1.2.14	Yes
	ACSL, fad; long-chain acyl-CoA synthetase	6.2.1.3	Yes
Fatty acid degradation	ACAT, atoB; acetyl-CoA C-acetyltransferase	2.3.1.9	Yes
	fadA, fadI; acetyl-CoA acyltransferase	2.3.1.16	Yes
	fadB; 3-hydroxyacyl-CoA dehydrogenase/enoyl-CoA hydratase/3-hydroxybutyryl-CoA epimerase/enoyl-CoA isomerase	1.1.1.35	Yes
	fadJ; 3-hydroxyacyl-CoA dehydrogenase/enoyl-CoA hydratase/3-hydroxybutyryl-CoA epimerase	1.1.1.35	Yes
	HAH; 3-hydroxyacyl-CoA dehydrogenase	1.1.1.35	Yes
	HAHA; enoyl-CoA hydratase/long-chain 3-hydroxyacyl-CoA dehydrogenase	4.2.1.17	No
	E1.3.3.6, ACOX1, ACOX3; acyl-CoA oxidase	1.3.3.6	No
	ACAS, bcd; butyryl-CoA dehydrogenase	1.3.8.1	No
	ACAM, acd; acyl-CoA dehydrogenase	1.3.8.7	No
	ACAL; long-chain-acyl-CoA dehydrogenase	1.3.8.8	No
	fadE; acyl-CoA dehydrogenase	1.3.99.-	No
	ACASB; short-chain 2-methylacyl-CoA dehydrogenase	1.3.8.5	No
	ACAVL; very long chain acyl-CoA dehydrogenase	1.3.8.9	No
	GCH, gcdH; glutaryl-CoA dehydrogenase	1.3.8.6	No
	ACSL, fad; long-chain acyl-CoA synthetase	6.2.1.3	Yes
	CPT1A; carnitine O-palmitoyltransferase 1, liver isoform	2.3.1.21	No
	ECI1, CI; elta3-elta2-enoyl-CoA isomerase	5.3.3.8	No
	alkB1_2, alkM; alkane 1-monooxygenase	1.14.15.3	No
	hca; 3-phenylpropionate/trans-cinnamate dioxygenase ferredoxin reductase component	1.18.1.3	No
	rubB, alkT; rubredoxin---NA+ reductase	1.18.1.1	No
	AH1_7; alcohol dehydrogenase 1/7	1.1.1.1	Yes
	frmA, AH5, adhC; S-(hydroxymethyl)glutathione dehydrogenase/alcohol dehydrogenase	1.1.1.284	No
	AH6; alcohol dehydrogenase 6	1.1.1.1	Yes
	adhE; acetaldehyde dehydrogenase/alcohol dehydrogenase	1.2.1.10	No
	ALH; aldehyde dehydrogenase (NA+)	1.2.1.3	Yes
	ALH7A1; aldehyde dehydrogenase family 7 member A1	1.2.1.31	No
	ALH9A1; aldehyde dehydrogenase family 9 member A1	1.2.1.47	No
	cyp_E, CYP102A, CYP505; cytochrome P450/NAPH-cytochrome P450 reductase	1.14.14.1	No
Fatty acid elongation	HAHB; acetyl-CoA acyltransferase	2.3.1.16	Yes
	HAH; 3-hydroxyacyl-CoA dehydrogenase	1.1.1.35	Yes
	ECHS1; enoyl-CoA hydratase	4.2.1.17	No
	PPT; palmitoyl-protein thioesterase	3.1.2.22	Yes
	ELOVL1; elongation of very long chain fatty acids protein 1	2.3.1.199	No
	HS17B12, KAR, IFA38; 17beta-estradiol 17-dehydrogenase/very-long-chain 3-oxoacyl-CoA reductase	1.1.1.62	No
	HAC, PHS1, PAS2; very-long-chain (3R)-3-hydroxyacyl-CoA dehydratase	4.2.1.134	No
	TER, TSC13, CER10; very-long-chain enoyl-CoA reductase	1.3.1.93	No
	ACOT1_2_4; acyl-coenzyme A thioesterase 1/2/4	3.1.2.2	Yes
Phenylalanine, tyrosine and tryptophan biosynthesis	E2.5.1.54, aroF, aroG, aroH; 3-deoxy-7-phosphoheptulonate synthase	2.5.1.54	Yes
	ARO1; pentafunctional AROM polypeptide	4.2.3.4	Yes
	aroKB; shikimate kinase/3-dehydroquinate synthase	2.7.1.71	Yes
	K16305; fructose-bisphosphate aldolase/6-deoxy-5-ketofructose 1-phosphate synthase	4.1.2.13	Yes
	K11646; 3-dehydroquinate synthase II	1.4.1.24	No
	aro; 3-dehydroquinate dehydratase I	4.2.1.10	Yes
	QUIB, qa-3; quinate dehydrogenase	1.1.1.24	No
	aroE; shikimate dehydrogenase	1.1.1.25	Yes
	quiA; quinate dehydrogenase (quinone)	1.1.5.8	No
	ydiB; quinate/shikimate dehydrogenase	1.1.1.282	Yes
	aroK, aroL; shikimate kinase	2.7.1.71	Yes
	aroA; 3-phosphoshikimate 1-carboxyvinyltransferase	2.5.1.19	Yes
	K24018; cyclohexadieny/prephenate dehydrogenase/3-phosphoshikimate 1-carboxyvinyltransferase	1.3.1.43	No
	aroC; chorismate synthase	4.2.3.5	Yes
	TRP3; anthranilate synthase/indole-3-glycerol phosphate synthase	4.1.3.27	Yes
	trp; anthranilate phosphoribosyltransferase	2.4.2.18	Yes
	trpF; phosphoribosylanthranilate isomerase	5.3.1.24	Yes
	priA; phosphoribosyl isomerase A	5.3.1.16	Yes
	trpC; indole-3-glycerol phosphate synthase	4.1.1.48	Yes
	TRP; tryptophan synthase	4.2.1.20	Yes
	E5.4.99.5; chorismate mutase	5.4.99.5	Yes
	tyrA1; chorismate mutase	5.4.99.5	Yes
	tyrA; chorismate mutase/prephenate dehydrogenase	5.4.99.5	Yes
	pheA1; chorismate mutase	5.4.99.5	Yes
	pheA; chorismate mutase/prephenate dehydratase	5.4.99.5	Yes
	AROA1, aroA; chorismate mutase	5.4.99.5	Yes
	aroH; chorismate mutase	5.4.99.5	Yes
	pheB; chorismate mutase	5.4.99.5	Yes
	tyrA2; prephenate dehydrogenase	1.3.1.12	No
	TYR1; prephenate dehydrogenase (NAP+)	1.3.1.13	No
	tyrC; cyclohexadieny/prephenate dehydrogenase	1.3.1.43	No
	tyrAa; arogenate dehydrogenase (NAP+)	1.3.1.78	Yes
	pheC; cyclohexadienyl dehydratase	4.2.1.51	Yes
	AT, PT; arogenate/prephenate dehydratase	4.2.1.91	Yes
	GOT1; aspartate aminotransferase, cytoplasmic	2.6.1.1	Yes
	TAT; tyrosine aminotransferase	2.6.1.5	Yes
	hisC; histidinol-phosphate aminotransferase	2.6.1.9	Yes
	tyrB; aromatic-amino-acid transaminase	2.6.1.57	Yes
	ARO8; aromatic amino acid aminotransferase I/2-aminoadipate transaminase	2.6.1.57	Yes
	ARO9; aromatic amino acid aminotransferase II	2.6.1.58	Yes
	pdh; phenylalanine dehydrogenase	1.4.1.20	No
	IL4I1; L-amino-acid oxidase	1.4.3.2	No
	phhA, PAH; phenylalanine-4-hydroxylase	1.14.16.1	No
	hphA; benzylmalate synthase	2.3.3.-	No
	hphC; 3-benzylmalate isomerase	4.2.1.-	No
	hphB; 3-benzylmalate dehydrogenase	1.1.1.-	Yes
	xanB2; chorismate lyase/3-hydroxybenzoate synthase	4.1.3.40	No
	fkbO, rapK; chorismatase	3.3.2.13	No
Terpenoid backbone biosynthesis	dxs; 1-deoxy--xylulose-5-phosphate synthase	2.2.1.7	Yes
	dxr; 1-deoxy--xylulose-5-phosphate reductoisomerase	1.1.1.267	Yes
	isp; 2-C-methyl--erythritol 4-phosphate cytidylyltransferase	2.7.7.60	Yes
	ispE; 4-diphosphocytidyl-2-C-methyl--erythritol kinase	2.7.1.148	Yes
	ispF; 2-C-methyl--erythritol 2,4-cyclodiphosphate synthase	4.6.1.12	Yes
	gcpE, ispG; (E)-4-hydroxy-3-methylbut-2-enyl-diphosphate synthase	1.17.7.1	Yes
	ispH, lytB; 4-hydroxy-3-methylbut-2-en-1-yl diphosphate reductase	1.17.7.4	No
	ACAT, atoB; acetyl-CoA C-acetyltransferase	2.3.1.9	Yes
	HMGCS; hydroxymethylglutaryl-CoA synthase	2.3.3.10	Yes
	HMGCR; hydroxymethylglutaryl-CoA reductase (NAPH)	1.1.1.34	Yes
	mvaA; hydroxymethylglutaryl-CoA reductase	1.1.1.88	No
	MVK, mvaK1; mevalonate kinase	2.7.1.36	Yes
	E2.7.4.2, mvaK2; phosphomevalonate kinase	2.7.4.2	Yes
	PMVK; phosphomevalonate kinase	2.7.4.2	Yes
	MV, mva; diphosphomevalonate decarboxylase	4.1.1.33	Yes
	pmd; phosphomevalonate decarboxylase	4.1.1.99	No
	ipk; isopentenyl phosphate kinase	2.7.4.26	No
	acnX1; mevalonate 5-phosphate dehydratase large subunit	4.2.1.-	No
	K25518; trans-anhydromevalonate 5-phosphate decarboxylase	4.1.1.-	Yes
	ubiX, bsdB, PA1; flavin prenyltransferase	2.5.1.129	No
	E2.7.1.185; mevalonate-3-kinase	2.7.1.185	No
	E2.7.1.186; mevalonate-3-phosphate-5-kinase	2.7.1.186	No
	E4.1.1.110; bisphosphomevalonate decarboxylase	4.1.1.110	No
	idi, II; isopentenyl-diphosphate elta-isomerase	5.3.3.2	Yes
	FPS; farnesyl diphosphate synthase	2.5.1.1	Yes
	E2.5.1.68; short-chain Z-isoprenyl diphosphate synthase	2.5.1.68	No
	ZFPS; (2Z,6Z)-farnesyl diphosphate synthase	2.5.1.92	No
	E2.5.1.86; trans, polycis-decaprenyl diphosphate synthase	2.5.1.86	No
	E2.5.1.88; trans, polycis-polyprenyl diphosphate synthase	2.5.1.88	No
	hexPS, COQ1; hexaprenyl-diphosphate synthase	2.5.1.82	No
	hexs-a; hexaprenyl-diphosphate synthase small subunit	2.5.1.83	No
	hepS; heptaprenyl diphosphate synthase component 1	2.5.1.30	Yes
	ispB; octaprenyl-diphosphate synthase	2.5.1.90	No
	SPS, sds; all-trans-nonaprenyl-diphosphate synthase	2.5.1.84	Yes
	PSS1; decaprenyl-diphosphate synthase subunit 1	2.5.1.91	No
	uppS; undecaprenyl diphosphate synthase	2.5.1.31	No
	NUS1; dehydrodolichyl diphosphate syntase complex subunit NUS1	2.5.1.87	No
	uppS, cpdS; tritrans, polycis-undecaprenyl-diphosphate synthase [geranylgeranyl-diphosphate specific		Yes
	chlP, bchP; geranylgeranyl diphosphate/geranylgeranyl-bacteriochlorophyllide a reductase	1.3.1.83	No
	ispS; isoprene synthase	4.2.3.27	No
	FNTA; protein farnesyltransferase/geranylgeranyltransferase type-1 subunit alpha	2.5.1.58	No
	RCE1, FACE2; prenyl protein peptidase	3.4.22.-	No
	STE24; STE24 endopeptidase	3.4.24.84	No
	ICMT, STE14; protein-S-isoprenylcysteine O-methyltransferase	2.1.1.100	No
	PCME; prenylcysteine alpha-carboxyl methylesterase	3.1.1.-	No
	PCYOX1, FCLY; prenylcysteine oxidase/farnesylcysteine lyase	1.8.3.5	No
	FOHSR; NAP+-dependent farnesol dehydrogenase	1.1.1.216	No
	FLH; NA+-dependent farnesol dehydrogenase	1.1.1.354	No
	FOLK; farnesol kinase	2.7.1.216	No
	K15793; acyclic sesquiterpene synthase	4.2.3.49	No

## Data Availability

Data is contained within the article or Appendix A. The data presented in this study are available.

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
