# Peer review of "Genome-Scale Metabolic Reconstruction, Non-Targeted LC-QTOF-MS Based Metabolomics Data, and Evaluation of Anticancer Activity of Cannabis sativa Leaf Extracts"

_metabolites, 2023, doi:10.3390/metabo13070788_

Round 1

Reviewer 1 Report

The manuscript provides plenty of information, especially concluded from Genome-Scale Metabolic Reconstruction, which I find described well in the manuscript. However, the experimental part of the manuscript is also relevant. I have found a few issues that should be corrected or presented with more detail to assure good comprehension of the paper. The details are given in the attached file.

Author Response

Dear Reviewer,

We would like to thank you for your valuable feedback and suggestions, which have greatly contributed to the improvement of our work. We have carefully considered your suggestions and have made several revisions.

We kindly request that you reconsider our revised manuscript.

Once again, we would like to express our gratitude for your valuable input and the time dedicated to reviewing our manuscript. We appreciate their expertise and constructive comments.

Best Regards

Reviewer 2 Report

At first the manuscript looked very promising. However there are some major issues that must be addressed before reconsideration of this manuscript will be made.

Abstract is a copy almost 1:1 from introduction which is highly unacceptable. Even bibliography is copied. I do not know if it is an error or purpose acting, here is some examples:

Abstract: (1) Colombia has suffered for the past decades a complex social problem related to illicit crops, including forced displacement, violence, and environmental damage, among other consequences for vulnerable populations [1].

Introduction: (1) Colombia has suffered for the past decades a complex social problem related to illicit crops, including forced displacement, violence, and and environmental damage, among other consequences for vulnerable populations [1].

(2) Here we present a Genome-Scale Metabolic network model of C. sativa with the analysis of non-targeted LC-MS based metabolomics data and evaluation of cytotoxicity and anticancer activity, which could help to understand their complex interactions toward applications in the anticancer, analgesic, and anti-inflammatory drug development

(2) Here we present a Genome-Scale Metabolic (GEM) reconstruction of C. sativa with the analysis of non- targeted LC-MS-based metabolomics data and evaluation of cytotoxicity and anticancer activity of leaf extracts, which could help to pave the way toward the discovery of anticancer, analgesic, and anti-inflammatory compounds

Abstract is shortened version of introduction. Must be rewritten in appropriate manner. At this point the manuscript is missing its crucial part. Among these there are other concerns that has to be addressed.

Little is presented in the introduction why the Authors input in the field is important and innovative. Why Authors used LC-MS? What what the reason behind it? Other methods compared to this one? Similar to abstract the introduction must be substantially rewritten to highlight why the finding are important and why such techniques are used. Are Authors sure that the LC-MS covered in introduction is the right equipment/instrument? I believe the LC-QTOF was used.

2.2.1 Sample prep. How this - the most important step of sample preparation was optimized? What conditions were checked and covered? How Authors are sure that the recoveries, transfer and recognition of analytes are covered by simple extraction? The three point calibration curve was used. Why 3 point cal curve? Again, how Authors optimized and chose the method for UPLC determination with PDA? This sentence is confusing: “The mobile phase used was 41:59 trichloroacetic acid 0.1% in water and acetonitrile.” Little is known about necessary steps, reproducibility, precision and accuracy of the method. Still unsure why the LC-PAD determination is under sample preparation.

2.2.2 Similar to previous paragraph, little is known about the optimization procedure. What kind of instrument was used as QTOF? 6540? This instrument does not allow to monitor in the same time in positive and negative mode. How this issue was resolved? By separate injections? Window mode analysis? Sequential experiments? Please explain. With all the respect, the eV are not used in the collision cell. How the equation was established? I am surprised that Authors uses equation while they are basing on many online databases where 0, 10, 20 and 40 V are used in collision cell at constant ratios – hence such fragmentation patterns and spectra are collected.

2.2.4 Again, as in whole manuscript: what assumptions led to the choice of such parameters? This is one the reason why the manuscript is not holding together sufficiently. Many parameters are introduced without proper explanation and basics.

At this point there is no reason to go through results, since the input parameters are not explained sufficiently. This can be done after addressing all the comments and possible resubmission.

English is fine

Author Response

Dear Reviewer,

We would like to thank you for your valuable feedback and suggestions, which have greatly contributed to the improvement of our work. We have carefully considered your suggestions and have made several revisions.

We kindly request that you reconsider our revised manuscript.

Once again, we would like to express our gratitude for your valuable input and the time dedicated to reviewing our manuscript. We appreciate your expertise and constructive comments.

Best Regards

Reviewer 3 Report

This is a very interesting paper on the genome-scale metabolic reconstruction of Cannabis sativa leaves and its medicinal properties. Extensive analyses were performed on the biomass to obtain the metabolites present, and to assess the cytotoxicity, and anticancer activity. Also, extensive metabolic pathways were identified for the specie’s metabolites.  

Overall, the paper is of good quality and results are robust. 

Therefore, I believe that the paper may to be published, but still I have some comments and revisions need to be made. 

Comments are made below, and on the paper, to improve the text:

Abstract

Remove references from abstract.

The abstract must include main results and conclusions, so I advise the authors to rephrase the abstract accordingly.

Introduction

Line 59-61 – what is the problem addressed? I understand that the plant is very interesting as medicinal, and this section provides sufficient and clear information on the subject. But how can research and development help to prevent illegal use of the plant?

Do you mean “to diversify and increase its value chain”? (line 381)

Material and methods

I guess most of the tools used are not free and require a license of use. So, the licenses number should be stated.

Number of samples performed in the assays is missing. 

Results:

Line 262 – please correct this caption from “Table” to “Figure”

Obviously there seems to be a clear correlation between the metabolites and the chemical reactions, and that is what the authors are looking for, but why isn’t this figure explored in the text? If it is superfluous, then put it in the supplement material. 

Line 271

Caption: 

C. satina in italic 

Also, explain what Rxns (reactions) and Cpds (compounds) are

Line 306 – correct the caption of Figure 4 (it is a bibliographic reference).

So, is this figure the authors interpretation of the metabolic pathways found in the present study?

Line 314, 331. It is not clear why there is a reference to Supplementary material here. That is, explain what is in the SM. 

Line 314. Explain in the caption what ESI mean (positive ionization, neutral mobile phase with negative ionization ESI, …..)

 Line 366. A statistical analysis is missing to assess if there are differences between healthy and cancer cells. It seems that healthy VERO cells are the most affected which is worrying. 

Discussion

Lines 182-183 and 401-405 – join the information from the two paragraphs below.

Line 544. I would be careful with the conclusions drawn regarding cytotoxicity, for the same effect was observed for VERO healthy cells. 

The authors cannot ignore this and so this section is misleading and must be reshaped. 

Conclusions are missing: what are the main conclusions and how is the “problem” answered by this paper?

The text is fairly good, although some editing is required, for I found some typo mistakes.

Author Response

(The authors gave the same response as above.)

Reviewer 4 Report

Here are my comments:

1.     Introduction: When the contingency caused by the coronavirus began, the former Minister of Health authorized Resolution 315 of 2020, which gives free rein to the sale of master formulas, which are preparations made under medical indication. In addition, two year later, Resolution 227 of 2022 was approved, regulating the use of use of medicinal C.sativa (non-psychoactive components) in food, beverages and dietary supplements. These new laws bring new great challenges and alternatives for the recovery of C.sativa products and their derivatives [7]. Additionally, since the beginning of this year, the national government through resolution 2808 of 2022 decided to include the magistral preparations of (C.sativa medicines) within the health benefits plan for patients with pathologies such as refractory epilepsy, fibromyalgia, sleep and appetite disorder, cachexia due to cancer, insomnia, chronic pain, neuropathic pain, and pain associated with cancer [8].

This paragraph is long & unnecessary.

2.     Introduction: Here we present a Genome-Scale Metabolic (GEM) reconstruction of C. sativa with the analysis of non- targeted LC-MS-based metabolomics data and evaluation of cytotoxicity and anticancer activity of leaf extracts.

What is LC-MS? As far as I know, it is liquid chromatography - mass spectrometry.

3.     Results: The PlantSEED semi-automatic reconstruction strategy was performed and curated with an exhaustive review of the literature and BLASTKOALA, to obtain the first C.sativa GEM reported in the literature (Figure 1).

What is The PlantSEED semi-automatic reconstruction strategy? Authors should have a brief introduction about it.

4.     Results: Un-targeted LC-MS based metabolomics data.

What does “un-targeted” mean? Why authors need “un-targeted” instead of “targeted”?

5.     Discussion: Finally, the notable activity against melanoma cells combined with the negligible impact against healthy human skin cells, confirms their great pharmacological potential that makes them suitable candidates for the development of new-generation topical treatments and specially for melanoma the more common and aggressive type of skin cancer with reduced side effects.

At what concentration C. Sativa is safe for healthy human skin cells? At that concentration, was C. Sativa addictive?

6.     Conclusion:

Add conclusion.

Minor editing of English language required

Author Response

(The authors gave the same response as above.)

Round 2

Reviewer 1 Report

I believe the revised version of the manuscript can be considered for publication. The Authors addressed most of my remarks. However, I would like to present a few suggestions regarding the proposed manuscript:

1.      Lines 261-262: The Authors did not specify what "high cannabidiol content" is. I suggest giving a specific value (in % or "more than ....%"), especially considering the deletion of the section presenting CDB, CBDA, and THC content in the samples in the revised text.

2.      Line 626: “Data extracted from LC-QTOF-MSLC-MS described some of the unique metabolites  of the species (cannflavin A) Table 34.” – the correction is not valid in my opinion. Please correct the number of Table and add a linking phrase or hyphen (at least)

3.      Line 644: "x" in the corrected word “their” should be removed

4.      Line 698: The sentence should start with “The notable….” – please verify the correction introduced.

minor typos

Author Response

Dear reviewer, 

We sincerely thank you for your comments and suggestions. We are confident that they have improved the quality of the manuscript and provided greater clarity.

Best Regards

Reviewer 2 Report

Dear Authors,

Thank You for improving Your manuscript. I believe now it is suitable for publication.

Author Response

Dear Reviewer, 

We sincerely thank you for your comments and suggestions. We are confident that they have improved the quality of the manuscript and provided greater clarity.

Best regards

Reviewer 4 Report

No comments

No comments

Author Response

(The authors gave the same response as above.)
